# Finite Element Simulation of the Flat Crush Behavior of Corrugated Packages

Jong-Min Park [1], Jae-Min Sim [2] and Hyun-Mo Jung [3,*]

1 Department of Bio-industrial Machinery Engineering, Pusan National University, Miryang 50363, Korea; parkjssy@pusan.ac.kr
2 Department of Digital Agriculture, Digital Agriculture Dissemination Team, Foundation of Agricultural Technology Commercialization & Transfer, Iksan 54667, Korea; woals447447@efact.or.kr
3 Department of Logistics Packaging, Kyongbuk Science College, Chilgok 39913, Korea
* Correspondence: hmjung@kbsc.ac.kr

**Abstract:** Corrugated paperboards are used for packaging because of their high strength-to-weight ratio, recyclability, and biodegradability. Corrugated paperboard consists of a liner and a corrugated medium and has an orthotropic sandwich structure with unique characteristics for each direction owing to its flute shape. In this study, finite element analysis (FEA) was performed on the flat crush behavior of the corrugated paperboard based on the flute type. The stress-strain (SS) curve and shape change of the flute were analyzed during the flat compression. In addition, it was compared with the FEA results through various experiments. The restraints and boundary conditions applied during FEA were used to properly describe the conditions during the experiment. Specifically, the horizontal translation motion of the top and bottom surfaces of the modeled test specimen was constrained during FEA to correspond to the effect of sandpaper attached to the upper and lower plates of the testing machine. This was done to prevent the specimen from sliding in one direction during the flat crush test. The change in the flute shape of the corrugated paperboard by flute type analyzed through experiments and FEA was very similar; although there was a difference in the absolute value between the two methods of the SS curve, the flute type exhibited a similar trend. Therefore, a qualitative comparative study on the flat crush behavior by flute type was possible with the FEA method, as in this study. Further studies on the material properties of the corrugated paperboard components and the modeling methods of the corrugated paperboard will enable the FEA-based simulation technique to be an alternative tool that can replace the flat crush test.

**Keywords:** corrugated package; finite element analysis; flat crush resistance; packaging material

## 1. Introduction

Recently, with emerging environmental problems worldwide, paper-based cushioning materials that can replace polymer-based ones, such as EPS (Expanded Polystyrene), EPE (Expanded Polyethylene), and EPU (Expanded Polyurethane), have gained interest in the packaging field.

The most representative paper-based packaging material is the corrugated paperboard, which is an orthotropic engineering structure. The mechanical and directional properties of the corrugated paperboard depend on its structural shape and the properties of the constituent paperboards [1–3].

Until recently, experimental methods have been primarily applied to the design problems and used for characteristic understanding of the various corrugated packages based on corrugated paperboards. However, these methods have limitations from both time and economical perspectives. In addition, it is impossible to quantitatively analyze the effect of each component of the corrugated paperboard on the mechanical strengths, such as the edgewise compression strength and flat crush resistance. Therefore, several

recent studies have applied finite element analysis (FEA) to the design of corrugated packages [2–9].

Most FEA studies have focused on analyzing the mechanisms of buckling, failure, stability, and adherence strength of the corrugated paperboard [4–6,10,11], and investigating the compression and flexural behavior of the different flute types [7]. Some FEA studies [8,12,13] have also attempted to replace the conventional semi-empirical equations by optimizing the board combinations of corrugated paperboard to analyze the performance and strength of their boxes and to develop FEA simulations. In addition, some researchers [3,8] conducted FEA on the various specimens of standard for the edgewise compression strength of corrugated paperboard worldwide, and compared and analyzed their results according to the specimen shape. Jiménez and Liarte [8] conducted FEA for edgewise compression test (ECT) specimens based on the recommendations of the FEFCO No.8 testing standard [14]. They found that the difference in the ECT values according to the frictional contact conditions between the modeled test specimen and the rigid surface of the testing machine was less than 3% in A/F (A-flute) and C/F (C-flute), and approximately 15% in B/F (B-flute). Park et al. [3] analyzed the edgewise compression behaviors of DW (single-wall) (A/F and B/F) and DW (double-wall) (AB/F and BB/F) corrugated paperboards that are commonly used in South Korea [15] with three standardized methods (KS M 7063-1 method A [16], TAPPI T 838 [17], and FEFCO No.8 [15]), using FEA simulations and experimental methods. They reported that the edgewise compression strength obtained by the FEA and the experiment differed due to the contact condition between the liner and flute in the FE modeling of the corrugated paperboard. Moreover, the difference between the standards by the experiment and the FEA was qualitatively consistent.

In this study, FEA for flat crush behavior, which is its out-of-plane characteristic, was conducted for each of the two types of SW (A/F and B/F) and DW (AB/F and BB/F) corrugated paperboards, which are commonly used in Korea. The effects of each component of the corrugated paperboard on the flat crush behavior were analyzed qualitatively. In addition, the applicability of the FEA-based computer simulation was analyzed through empirical experiments on the FEA results.

## 2. Experiment Design

### 2.1. Flat Crush Test

Table 1 lists the four types of samples applied to the flat crush analysis of the corrugated paperboard, and the board combination and geometric dimensions of the target corrugated paperboards.

**Table 1.** Measured specifications of the corrugated paperboards used in the study.

| Kinds | | Board Combination | Flute | | | Paperboards |
|---|---|---|---|---|---|---|
| | | | Wave Length (λ) (mm) | Height (h) (mm) | Take-Up Factor | |
| SW | A/F | SK180/K180/SK180 | 9.00 (8.33–9.38) | 4.90 (4.5–4.8) | 1.560 (1.6) | - Thickness (mm): 0.22 (SK180). 0.24 (K180) |
| | B/F | SK180/K180/SK180 | 6.00 (5.27–6.25) | 2.65 (2.5–2.8) | 1.424 (1.4) | - Ring crush (kgf): 21.7 (SK180). 20.2 (K180) |
| DW | AB/F | SK180/K180/K180/K180/SK180 | - | - | - | - Tensile strength (MPa): 66.33 (MD)/22.76 (CD) (SK180), |
| | BB/F | SK180/K180/K180/K180/SK180 | - | - | - | 52.97 (MD)/18.18 (CD) (K180) |

Notes: ( ) KS T 1034 [18]; Total thickness (mm) of corrugated paperboard: (A/F) 5.34, (B/F) 3.09, (AB/F) 8.23, (BB/F) 5.98; K180; 100% KOCC, KOCC = Korean old corrugated container; SK180: 20% outer liner containing UKP + 80% KOCC, UKP = unbleached kraft pulp.

For the flat crush test, 10 round specimens of Φ90.6 mm were manufactured according to KS M ISO 3035 [19] for each type of sample as shown in Figure 1. The manufactured test specimens were sufficiently equilibrated for more than 48 h in a thermo-hygrostat under standard conditions (23 ± 2 °C, RH 50 ± 2%). Thereafter, the experiment was performed

in a laboratory where the temperature and relative humidity were maintained relatively well [20].

In the flat crush test of the corrugated paperboard, the loading rate was 12.5 ± 0.25 mm/min [19]. Sandpaper was attached to the upper and lower plates of the compression tester in contact with the test specimen to alleviate the experimental error induced when the test specimen is pushed to one side during compression.

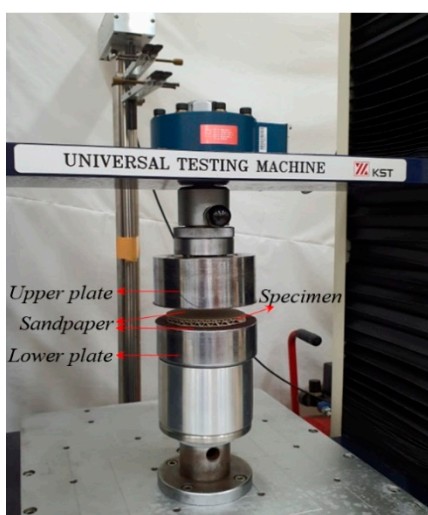

**Figure 1.** Flat crush test fixture of the corrugated paperboard.

### 2.2. FE Modeling and Analysis Procedures

The finite element (FE) model was created based on the physical specifications (Table 1) of the two SW (A/F and B/F) and two DW (AB/F and BB/F) corrugated paperboards used for the flat crush test. The geometrical shape of the flute was modeled as a cosine function. For simplification, the connection point between the liner and flute of the corrugated paperboard was modeled using a sharing method for the points (nodes).

In the FEA, the dimensions of the modeled test specimen were set to MD × CD = 36 × 9 mm, so that it was an integer multiple of 4 for A/F and 6 for B/F. These dimensions can improve convergence and reduce analysis errors that are caused by changes in the flute shape based on the contact conditions of the upper and lower flutes determined through preliminary analysis of DW-AB/F.

Figure 2 presents the FE model mesh and geometry of the target corrugated paperboards by flute type. In particular, in the case of DW-BB/F, which is increasingly being used in Korea, the case of a three-phase angle was modeled (Figure 3). The hexahedral mesh was used due to the shape of corrugated paperboard. The computational domains for A/F, B/F, AB/F-DW, and BB/F-DW were discretized into 144,228 nodes and 94,200 mesh elements, 143,252 nodes and 93,180 mesh elements, 156,825 nodes and 102,400 mesh elements, and 148,767 nodes and 97,200 mesh elements, respectively.

The post-processor used in this study was ANSYS® FE [21] using workstation computer (Z1 G6 8YH59AV, HP, Daejeon, Korea). Nonlinear/large displacement conditions were applied to the FEA owing to the material properties of the corrugated paperboard.

Frictional contact conditions were applied to the parts where the contact between the corrugated medium and the liner was expected when the FE model was compressed. The boundary and constraint conditions applied to the FE model in the FEA were set similar to those in the experiment. Thus, by constraining the bottom liner of the FE model with a fixed support, both the translation and rotation in the $x$, $y$, and $z$ directions were constrained. The horizontal translation movement ($x$ and $y$-direction in Figure 2) of the top liner of SW, along with the horizontal translation movement of both the top and middle liner of DW, were constrained.

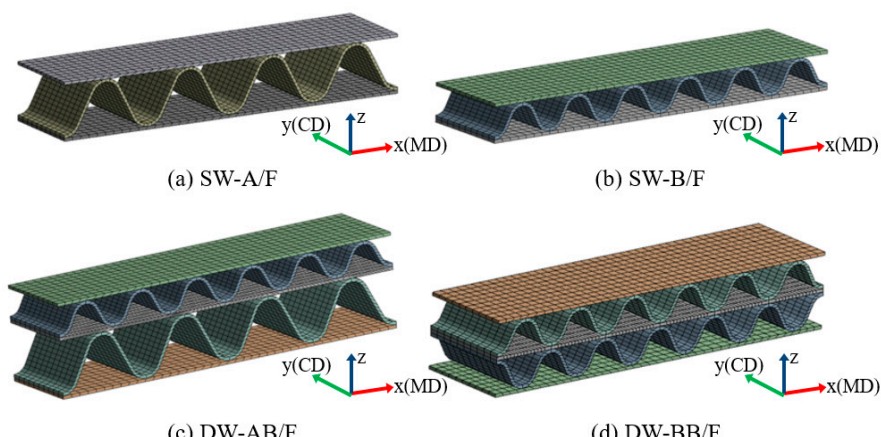

**Figure 2.** Meshed 3D models for FE simulation.

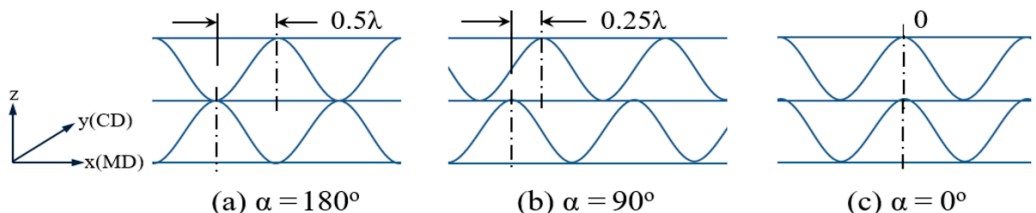

**Figure 3.** DW-BB/F of different phase angle (α) applied to the FEA. (**a**) the difference between the phase angle of the upper and lower flute is 180°, (**b**) the difference between the phase angle of the upper and lower flute is 90° and (**c**) the difference between the phase angle of the upper and lower flute is 0°.

To apply the load to the modeled test specimen, its top surface was displaced downward at a speed of 12.5 mm/min. Consequently, the deformation caused by the moving plates of the compression tester used in the experiment was simulated [19]. The reaction force applied to the bottom surface of the modeled test specimen was analyzed to obtain the load vs. displacement plots.

### 2.3. Material Properties

The required material properties for FEA were the Young's modulus, Poisson's ratio, shear modulus, yield strength, and frictional coefficient among the corrugated paperboard components. These components, such as liners and corrugated mediums, were assumed to be orthotropic materials [22], implying that their material properties are symmetrical for x(MD), y(CD), and the z-plane (Figure 2). Therefore, each paperboard possessed nine elastic material properties, such as $E_x$, $E_y$, $E_z$, $G_{xy}$, $G_{xz}$, $G_{yz}$, $\mu_{xy}$, $\mu_{xz}$, and $\mu_{yz}$, and two strength values, $\sigma_x$(MD) and $\sigma_y$(CD) [3,23]. The values of these properties in the FEA were set based on those reported by Park et al. [3], as listed in Table 2.

**Table 2.** Orthotropic material properties of corrugated paperboards used for the FEA [3].

| Paperboards | Young's Modulus (GPa) | | | Poisson's Ratio | | | Shear Modulus (GPa) | | | Yield Strength (MPa) | |
|---|---|---|---|---|---|---|---|---|---|---|---|
| | $E_x$ (MD) | $E_y$ (CD) | $E_z$ | $\mu_{xy}$ | $\mu_{xz}$ | $\mu_{yz}$ | $G_{xy}$ | $G_{xz}$ | $G_{yz}$ | $\sigma_x$ (MD) | $\sigma_y$ (CD) |
| K180 | 2.20 (±0.02) | 0.37 (±0.01) | 0.011 | 0.34 | 0.01 | 0.01 | 0.349 | 0.040 | 0.010 | 29.09 (±0.8) | 12.12 (±0.1) |
| SK180 | 3.16 (±0.07) | 0.40 (±0.01) | 0.016 | 0.34 | 0.01 | 0.01 | 0.435 | 0.057 | 0.011 | 42.50 (±0.8) | 19.50 (±0.5) |

Note: ( ) standard deviation.

As the modeled test specimen was compressed, the contact area between the flute and liner on the left and right portions of the shared nodes increased gradually. Accordingly, the frictional contact conditions at the portion where contact was expected were implemented to accurately analyze the flat crush behavior of the corrugated paperboard. In this study, the static-frictional coefficients, listed in Table 3, were set based on those reported by Park et al. [3].

**Table 3.** Frictional coefficients between paperboards used for the FEA [3].

| Classify | Static-Frictional Coefficient | | |
|---|---|---|---|
| | MD-MD | CD-CD | MD-CD |
| K180-K180 | 0.23 (±0.02) | 0.29 (±0.01) | 0.26 (±0.02) |
| SK180-SK180 | 0.37 (±0.06) | 0.41 (±0.03) | 0.39 (±0.03) |
| K180-SK180 | 0.23 (±0.04) | 0.35 (±0.02) | 0.32 (±0.04) |
| Average | 0.28 | 0.35 | 0.32 |

Note: ( ) standard deviation.

## 3. Results and Discussion

### 3.1. FE Simulation for Flat Crush Behavior

The flat crush process was analyzed by forcibly displacing the modeled test specimen for each flute type at a speed of 12.5 ± 2.5 mm/min, which is the flat crush test condition [19]. Figure 4 presents an example of the FEA results when 20% of the total thickness of the modeled test specimen for each flute type is compressed. Table 4 summarizes the FEA results by flute type and compression levels.

When the modeled test specimen was subjected to flat compression at a constant speed, in the case of SW (A/F and B/F), the shape of the flutes changed in the order of a half-sine wave, trapezoid, square, and Ω shape. Particularly, in the case of A/F, the bottom part of the flute spread more widely and contacted the liner; in the case of B/F, both the top and bottom parts of the flute contacted the liner at the same ratio. In the case of AB/F, the A/F deformed more and faster than B/F, regardless of the compression direction.

In the case of BB/F (Figure 3), the flute deformation process was clearly distinguished according to the phase difference between the upper and lower flutes. Thus, in the case of a phase difference of 180° (Figure 3a), the shape of the two flutes was maintained almost symmetrically, which was advantageous in terms of crush failure and flute recovery, compared to other phase differences under the same conditions. However, when BB/F has a phase difference in which neither the top nor the bottom part of the upper and lower flutes come into contact with each other, the change process of the two flutes is significantly different based on the middle liner. Consequently, the stiffness of the middle liner is the main variable in the flute deformation. Therefore, the case of a 180° phase difference is less affected by the middle liner. Accordingly, the edgewise compression strength of the DW corrugated paperboard is less affected by the middle liner than the flat crush resistance. According to Armentani et al. [2], the phase difference between two flutes (having the same flute type) of the DW corrugated paperboard did not significantly affect the buckling strength during edgewise compression.

Figure 5 shows the stress-strain curve (SS curve) of the modeled test specimen by flute type obtained using FEA. The values on the graph are converted to the size of a circular test specimen (Φ90.6 mm) in the experiment by considering the ratio of the dimensions of MD and CD acquired from the FEA result for the modeled test specimen (MD × CD = 36 × 9 mm).

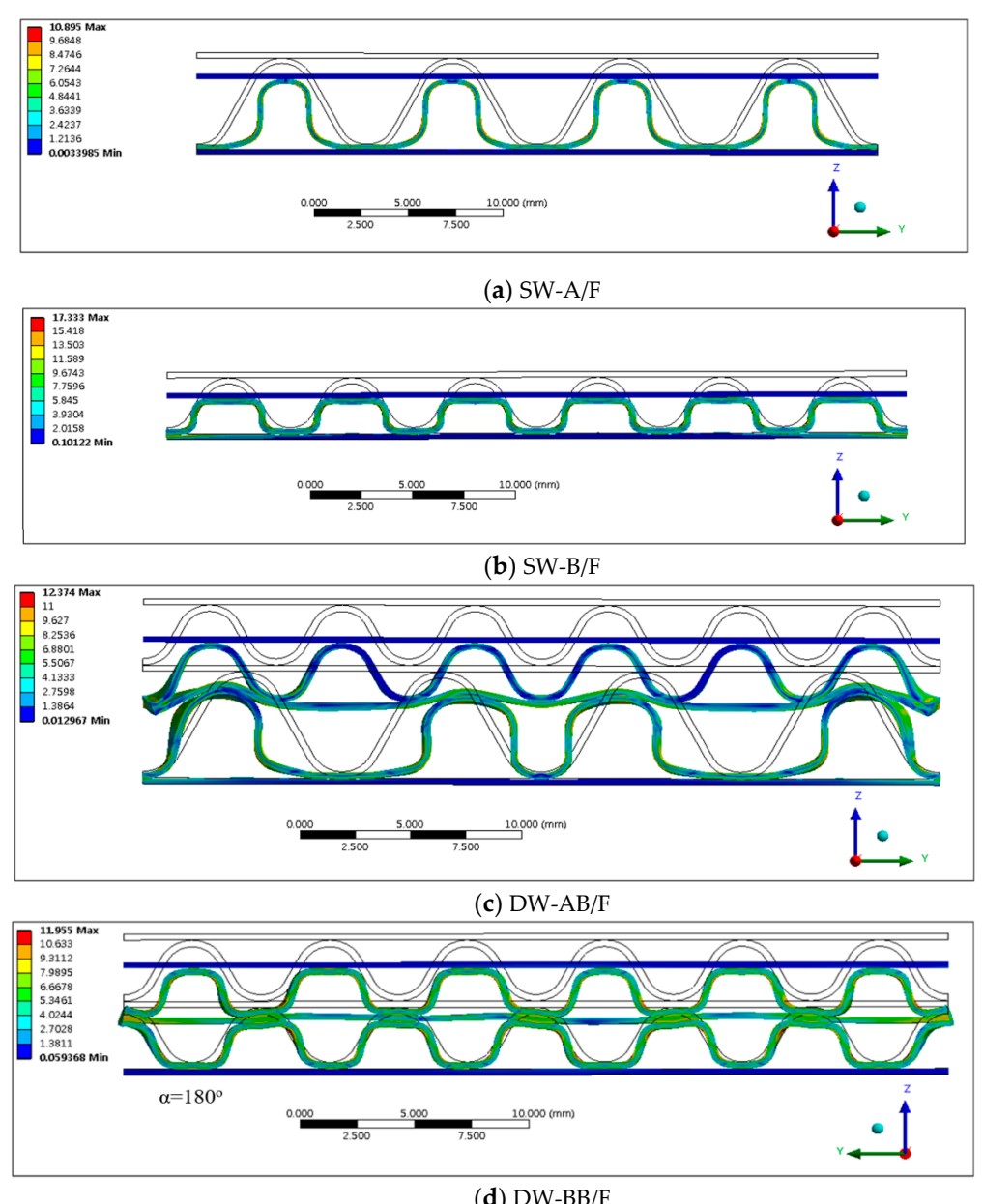

(**a**) SW-A/F

(**b**) SW-B/F

(**c**) DW-AB/F

(**d**) DW-BB/F

**Figure 4.** FEA result of the modeled test specimen by flute type (20% compression for each thickness of the modeled test specimen).

**Table 4.** Flat crush process of the modeled test specimen by flute type through FEA.

| Flute Type | Compression Degree for Total Thickness of the Modeled Test Specimen | | | | | |
| --- | --- | --- | --- | --- | --- | --- |
| | 0% | 10% | 20% | 30% | 40% | 50% |
| A/F | | | | | | |
| B/F | | | | | | |
| AB/F (B/F→A/F) | | | | | | |
| AB/F (A/F→B/F) | | | | | | |
| BB/F ($\alpha = 180°$) | | | | | | |

**Table 4.** *Cont.*

| Flute Type | Compression Degree for Total Thickness of the Modeled Test Specimen | | | | | |
|---|---|---|---|---|---|---|
| | 0% | 10% | 20% | 30% | 40% | 50% |
| BB/F ($\alpha = 90°$) | 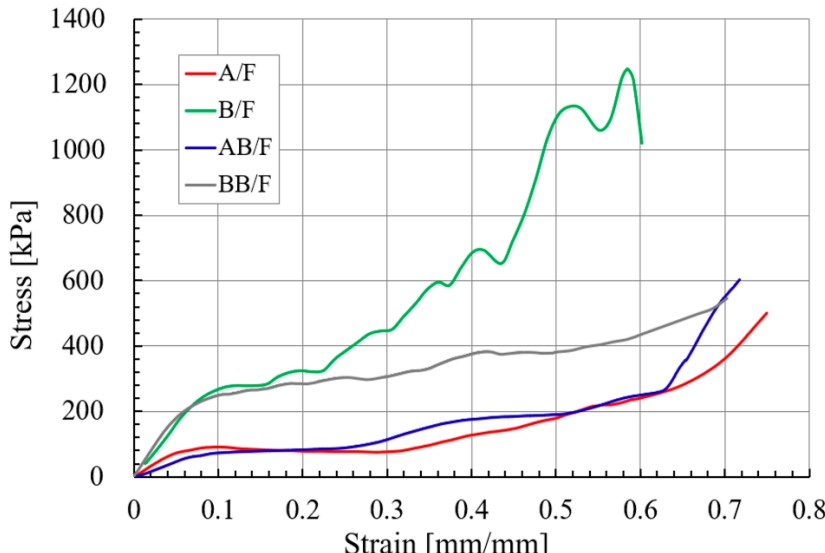 | | | | | |
| BB/F ($\alpha = 0°$) | | | | | | |

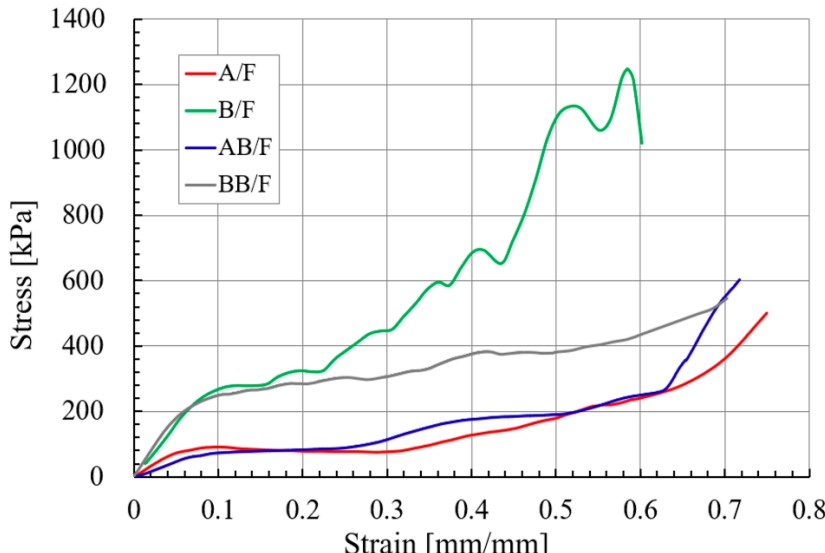

**Figure 5.** SS curves of the corrugated paperboard by flute type obtained using FEA.

The slope of the SS curve was significantly larger in B/F, followed by BB/F, A/F, and AB/F. Moreover, the difference between A/F and AB/F was very small. This phenomenon is opposite to the flexural behavior, and the resistance to flexural stiffness increases as the thickness of the corrugated paperboard increases under similar conditions [24]. However, in the case of flat crush behavior, the smaller the thickness and the more compact the structure, the greater the resistance to the flat crush (flat crush stiffness) [24].

Figure 6 demonstrates the flat crush behavior according to the compression direction of DW-AB/F. Within a compressive strain of 60%, the difference between the two compression directions is minimal, but at a rate exceeding 60%, the resistance to flat crush in the case of the compression of B/F→A/F was greater than that of the opposite case. However, through a four-point flexural analysis for the same corrugated paperboards, Sim [24] reported that the resistance to flexural for B/F→A/F was greater than that for the opposite direction.

Figure 6 also demonstrates the flat crush behavior based on the phase difference between the upper and lower flutes of the DW-BB/F. Within 50% of the compressive strain, the case where the phase difference was 180° showed higher flat crush resistance than the other phase differences. This is because, in the case of a 180° phase difference, the top and bottom parts of the upper and lower flutes are in direct contact with each other to maintain symmetrical deformation. However, the SS curve or the flexural stiffness exhibited no difference owing to the mismatch between the upper and lower flutes, as indicated in the FEA results of the flexural behavior for the same material [24].

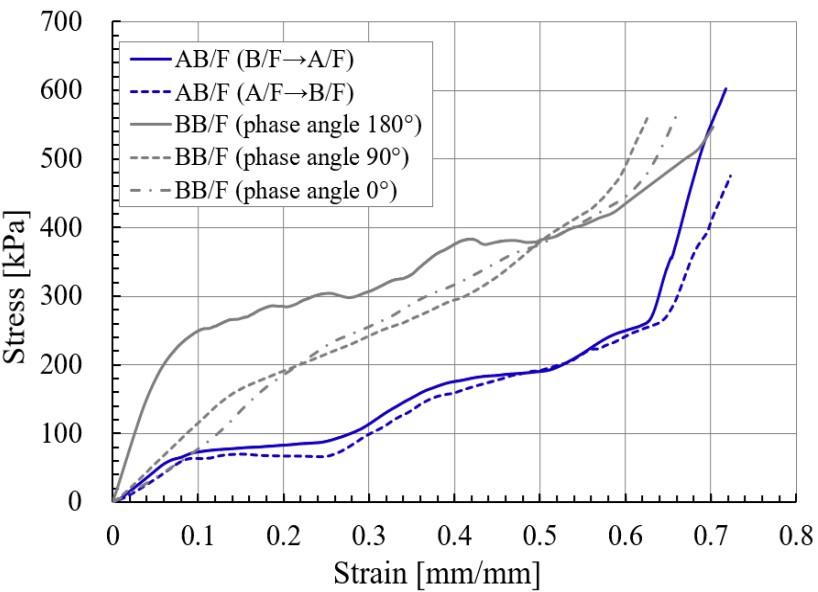

**Figure 6.** SS curves of the DW corrugated paperboard obtained using FEA.

*3.2. Comparison with the Experimental Study*

Figures 7 and 8 present the SS curves of the corrugated paperboard (circular specimens of Φ90.6 mm [19]) by flute type analyzed through experiments, as well as using FEA.

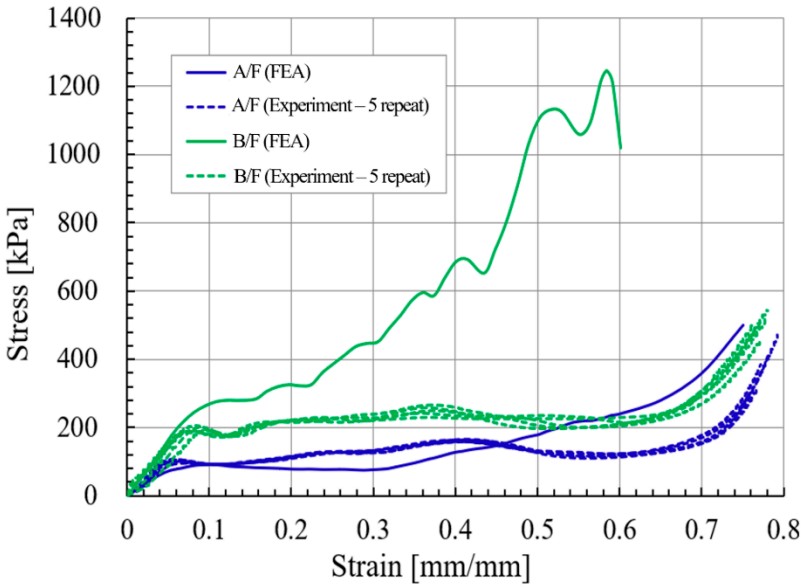

**Figure 7.** Comparison of the SS curves of the SW corrugated paperboard acquired through experiments and FEA.

In the case of SW-A/F, the SS curves obtained through the experiment and FEA were in good agreement within a compressive strain of approximately 45%. However, in the case of SW_B/F and DW_BB/F, the SS curves between the two methods of the experiment and FEA were well matched in the initial elastic region, but from a compressive strain of approximately 6.5% or more, the SS curve through the FEA increased more steeply than that through the experiment. The difference between the experimental and FEA results for each flute type can be substantially reduced by applying viscoelastic properties to the paperboard during FEA. However, the difference in the SS curves for each flute type by FEA shows the same trend as in the experiment. Therefore, a qualitative analysis of the flat crush behavior for each flute type is possible with the FEA method used in this study.

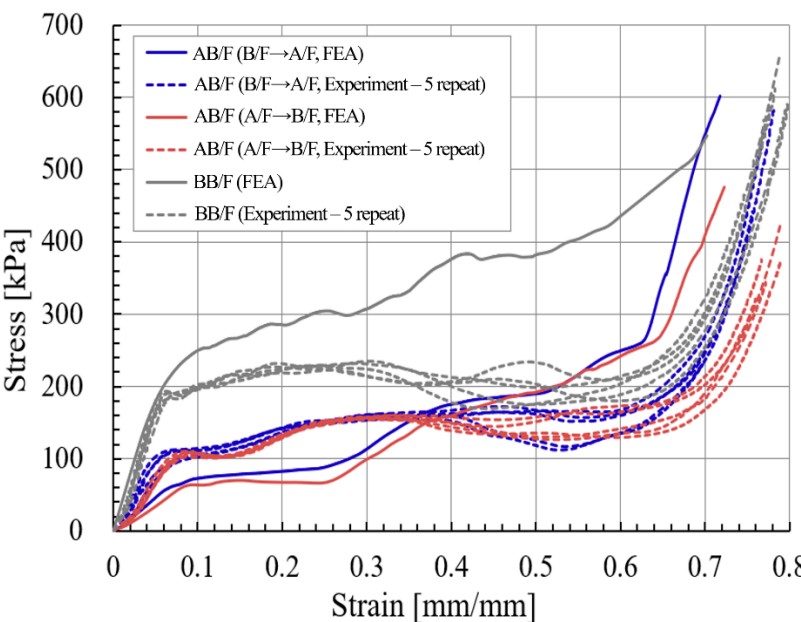

**Figure 8.** Comparison of the SS curves of the DW corrugated paperboard acquired through experiments and FEA.

As shown in Figure 8, which shows the SS behavior according to the compression direction of DW-AB/F, the difference in the compression direction obtained by the experiment was very similar to that obtained by the FEA. The compression direction between the two methods exhibited no difference within a compressive strain of 60%.

Table 5 presents the deformation process of the flute shape when the flat crush test was performed on the specimen of the corrugated paperboard. In the case of SW (A/F, B/F), the shape of the flute changed in the order of half-sine wave, trapezoid, square, and Ω shape, similar to the FEA results presented in Table 4. This is because the horizontal movement of the upper and lower surfaces of the specimen was suppressed by sandpaper during the experiment and by the restraints of the nodes during the FEA.

**Table 5.** Flat crush process of modelled test specimen by flute type through experiment.

| Flute Type | Compressive Degree of the Modeled Test Specimen | | | | | |
|---|---|---|---|---|---|---|
| | 1 | 2 | 3 | 4 | 5 | 6 |
| A/F<br>B/F | | | | | | |
| AB/F<br>(B/F→A/F) | | | | | | |
| AB/F<br>(A/F→B/F) | | | | | | |
| BB/F<br>(α = 180°) | | | | | | |

The flat crush behavior of the DW corrugated paperboard is shown by the interaction between two flutes, which are usually made of the same paperboard. In the case of AB/F, A/F with a large flute height and wavelength undergoes more deformation than B/F, and eventually collapses. Subsequently, the body loses resistance to compressive loads and undergoes plastic deformation. Therefore, the resistance of A/F determines the

total amount of flat crush deformation before the collapse of the DW-AB/F corrugated paperboard. Meanwhile, in the case of DW-BB/F with a phase difference of 180°, where the top and flute bottom of the upper and lower flutes are in direct contact, the two flutes are deformed almost simultaneously and support the deformation and load together, similar to the FEA results.

Tables 4 and 5 present FEA and experimental results, respectively, of the analysis with the constrained horizontal translation motion of the top and bottom surfaces of the (modeled) test specimen. If they were not constrained (i.e., sandpaper was not used for testing and friction contact conditions were provided for FEA), the specimens were compressed occasionally while being pushed in one direction. In that case, the experimental error was large, and the convergence was significantly lower in the FEA.

### 4. Conclusions

Corrugated paperboard is a representative of eco-friendly packaging material, which has a high strength-to-weight and stiffness-to-weight ratio compared to polymer-based cushioning materials. Therefore, the demand for corrugated paperboard-based packages as a substitute for polymer-based cushioning materials is expected to increase.

In this study, we analyzed the flat crush behavior of corrugated paperboard by flute type using FEA, as well as the similarity with the experimental results. Our results can be summarized as follows:

1. The experimental results confirmed that when the SW corrugated paperboard was flat-compressed, the shape of the flute changed in the order of half-sine wave, trapezoid, square, and character 'ohm' shape.
2. In both the experimental and FEA results of the flat crush behavior of the DW-AB/F corrugated paperboard, we found that A/F deformed more and faster than B/F to reach a state of collapse, regardless of the compression direction. Therefore, the effect of A/F on the resistance to flat crushing and deformation of the DW-AB/F corrugated paperboard was significant.
3. In the case of DW-BB/F, the phase difference between the upper and lower flutes had a significant effect on the variation in the flute shape. In the case of a phase difference of 180°, wherein the top and bottom of the upper and lower flutes coincide, the effect of the middle liner on the change of the flute shape was smaller than that in the case of other phase differences. This is because the two flutes changed symmetrically, and the resistance to flat crush was significant. In addition, differences in the SS curve according to the phase difference between the upper and lower flutes were observed. For a phase difference of 180°, the slope of the curve was larger within 50% of the compressive strain, compared to other phase differences.
4. The slope of the SS curve for flat crushing of the corrugated paperboard by flute type through FEA was significantly larger in B/F, followed by BB/F, A/F, and AB/F; moreover, the difference between A/F and AB/F was minimal. This tendency was also confirmed in the experimental results. However, there was an absolute difference in the SS curves obtained by FEA and experiment. The difference could be significantly reduced by applying the viscoelastic properties of the paperboard in the FEA.
5. If the material properties of the corrugated paperboard components and its modeling methods are further studied, the FE-based simulation technique will be a useful alternative tool that can replace the flat crush test.

**Author Contributions:** Data curation, J.-M.P. and H.-M.J.; Formal analysis, J.-M.P. and J.-M.S.; Investigation, H.-M.J.; Software, J.-M.S. and H.-M.J.; Validation, J.-M.P.; Visualization, J.-M.S.; Writing-original draft, J.-M.P.; Writing-review and editing, H.-M.J.; Supervision, H.-M.J. All authors have read and agreed to the published version of the manuscript.

**Funding:** This research received no external funding.

**Institutional Review Board Statement:** Not applicable.

**Informed Consent Statement:** Not applicable.

**Data Availability Statement:** Data available on request due to restrictions eg privacy or ethical.

**Conflicts of Interest:** The authors declare no conflict of interest.

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
