# Peer review of "Finite Element Simulation of the Flat Crush Behavior of Corrugated Packages"

_applsci, doi:10.3390/app11177867_

Round 1

Reviewer 1 Report

This is an interesting work regarding FEM simulation of corrugated packages. However various points need to be addressed. 

  • A nonlinear large displacement takes place for the simulations; however, it is not clear if it is a static or a transient analysis. How long is the simulation time if it is a transient analysis. Since no time is shown in the results, and no density is needed as a material property, I would assume it is static. If so, wouldn’t a transient analysis be more reasonable?
  • Moreover, details should be given for the number of elements used and what type of elements. Was a convergence analysis performed to end up to the final mesh? Furthermore, what is the duration of the simulation in real time and some specifications for the PC or cluster where the simulations run should be given.
  • Table 1 and 2 should be rewritten to correct some words that don’t fit in the same line such a take-up factor, DW, (mm), and some parentheses that are in the next line. Some unnecessary bold type words should be also omitted in Table 1.
  • A few more details should be given for the sharing method used for the liner and the flute and the contact conditions, such as number of shared nodes.
  • On page 3 in the last paragraph, figure 2 presents the mesh and geometry of the models not the modeling results. Correct this.
  • In figure 4 they should write in the legend that they show results for the DW-BB/F (α=180o), since the phase difference is not given.
  • The authors claim that in the case of SW (A/F and B/F), the shape of the flutes changed in the order of a half-sine wave, trapezoid, square, and Ω shape. Can they illustrate better this claim in either Table 4 or 5 by pointing out graphically these four shapes for the case of A/F or B/F.
  • The authors claim that “in the case of SW_B/F and DW_BB/F, the SS curves between the two methods were well matched in the initial elastic region, but from a compressive strain of approximately 6.5%, the SS curve through the FEA increased more steeply than that through the experiment” and they attribute this to improper use of material properties by suggesting to use viscoelastic properties in the future. This may be true however a transient analysis, with static and dynamic frictional coefficients, may also provide more accurate results, along with the use of an elasto-plastic strength material model.
  • In table 5 the B/F case is missing.
  • In figure 8 correct inside the graph the word exprei. to experi. Also what the word repli. means? This should also be written in the text. These are 5 different experimental cases that differ in what?
  • On page 7 the authors write: ”This phenomenon is opposite to the flexural behavior, and the resistance to flexural stiffness increases as the thickness of the corrugated paperboard increases under similar conditions [24]. However, in the case of flat crush behavior, the smaller the thickness and the more compact the structure, the greater the resistance to the flat crush (flat crush stiffness) [24].” They should better correlate these claims with their results, since it is unclear to the reader what they mean in relation to their results.

Reviewer 2 Report

Title: Finite Element Simulation of the Flat Crush Behavior of Corrugated Packages

Author: Jong Min Park , Jae Min Sim , Hyun Mo Jung

Manuscript ID: applsci-1344556

General comments:

The authors studied the flat crush behavior of corrugated packages using finite element method under compression. The research study is interesting and the manuscript is well written.

Specific comments:

  1. Line 17, “… empirical experiments.”, not clear about this.
  2. Line 34, expand the EPS, …, when they first appeared in the manuscript.
  3. Lines 117-118, “Frictional contact conditions … was compressed”, is there any contacted point slipped during the compression test?
  4. The present study used the frictional coefficient obtained from Ref. [3], is the present model same with the model used in Ref. [3], this might result in the difference in FEA and experimental result shown in Figure 7
  5. In the linear region in Figure 8, the effective slope between the test and FEA results is not close there might be some differences in the material properties. Is there any way to test a coupon to obtain the measured material property? After that, you may re-run the nonlinear static analysis to obtain the stress-strain relation.

Round 2

Reviewer 1 Report

I accept the manuscript in its current form.